# Reduced excitatory neuron activity and interneuron-type-specific deficits in a mouse model of Alzheimer's disease

Moustafa Algamal[1], Alyssa N. Russ[1], Morgan R. Miller[1], Steven S. Hou[1], Megi Maci[1], Leon P. Munting[1], Qiuchen Zhao[1], Dmitry Gerashchenko[2], Brian J. Bacskai[1✉] & Ksenia V. Kastanenka[1✉]

Alzheimer's disease (AD) is characterized by progressive memory loss and cognitive decline. These impairments correlate with early alterations in neuronal network activity in AD patients. Disruptions in the activity of individual neurons have been reported in mouse models of amyloidosis. However, the impact of amyloid pathology on the spontaneous activity of distinct neuronal types remains unexplored in vivo. Here we use in vivo calcium imaging with multiphoton microscopy to monitor and compare the activity of excitatory and two types of inhibitory interneurons in the cortices of APP/PS1 and control mice under isoflurane anesthesia. We also determine the relationship between amyloid accumulation and the deficits in spontaneous activity in APP/PS1 mice. We show that somatostatin-expressing (SOM) interneurons are hyperactive, while parvalbumin-expressing interneurons are hypoactive in APP/PS1 mice. Only SOM interneuron hyperactivity correlated with proximity to amyloid plaque. These inhibitory deficits were accompanied by decreased excitatory neuron activity in APP/PS1 mice. Our study identifies cell-specific neuronal firing deficits in APP/PS1 mice driven by amyloid pathology. These findings highlight the importance of addressing the complexity of neuron-specific deficits to ameliorate circuit dysfunction in Alzheimer's disease.

[1] Department of Neurology, MassGeneral Institute of Neurodegenerative Diseases, Massachusetts General Hospital and Harvard Medical School, Charlestown, MA, USA. [2] Harvard Medical School/VA Boston Healthcare System, West Roxbury, MA, USA. ✉email: BBACSKAI@mgh.harvard.edu; KKASTANENKA@mgh.harvard.edu

Alzheimer's disease (AD) is the most common form of dementia and the most prevalent neurodegenerative disorder in the United States[1]. The primary pathological hallmarks of Alzheimer's disease include deposits of extracellular amyloid plaques as well as intracellular tau tangles, and eventually neuronal death[2,3]. Notably, the accumulation of amyloid-beta peptides is associated with aberrant neuronal activity and oscillatory network alterations in AD patients[4] and in animal models[5–8]. Particularly, deficits in cognition-linked brain rhythms such as gamma and slow waves are prevalent in AD patients[9,10]. These neuronal activity alterations appear at the early stages of AD and correlate with the severity of cognitive impairment in AD patients[11,12]. Oscillatory brain rhythms are generated by complex firing patterns of individual neurons and interactions between neuronal populations across different brain regions[13–15]. Thus, there is an urgent need to understand the single-cell neuronal activity patterns in AD to restore the impaired oscillatory brain rhythms and improve cognitive function.

Several studies have assessed single-cell neuronal activity in amyloidosis mouse models in ex-vivo brain slices[16–19] and in vivo[5,20,21]. While ex-vivo studies have monitored cell-type-specific neuronal activity, in vivo studies did not discriminate between neuronal subtypes and mainly reported the activity of excitatory neurons, which represent 80–90% of cortical neurons[22]. These reports have provided mixed findings of aberrant neuronal hyperactivity[5,6] and hypoactivity[5,21].

In addition to excitatory cells, ~10–20% of cortical neurons are inhibitory interneurons that display a wide variety of morphological and electrophysiological properties[22–25]. Despite constituting a small percentage, interneurons play an essential role in cortical computations and the maintenance of spontaneous network oscillations[24]. Interneurons are classified based on their protein expression profiles into distinct subtypes including somatostatin (SOM)- and parvalbumin (PV)-expressing interneurons, as well as others[26]. SOM and PV interneurons shape excitatory activity by targeting different cellular compartments of pyramidal cells[24].

The function of specific interneuronal cell types is impaired in AD[27,28]. Particularly, PV interneuron dysfunction received considerable attention because of its role in gamma oscillatory activity[27,29]. For instance, hypoactivity[17,18,29–32] and hyperactivity[19] of PV interneurons were reported in several AD mouse models of amyloidosis. The firing activity of SOM interneurons in AD mouse models remains unexplored, but impairments in synaptic rewiring of hippocampal SOM interneurons was reported[28]. These cell-type-specific dysfunctions disrupt the well-regulated excitation and inhibition (E/I) balance resulting in broader, circuit-level dysfunction in AD[27]. Whether all interneuronal types are affected similarly in AD and whether their function is directly affected by amyloid plaques remains unclear.

During action potential firing, calcium ions ($Ca^{2+}$) enter neurons through voltage gated calcium channels resulting in intracellular $Ca^{2+}$ concentration changes[33]. As result, calcium imaging has been widely used as a proxy of neuronal action potential firing[20,34,35].

Here, we determined the effect of amyloid-beta accumulation on the spontaneous activity of three distinct neuronal populations within cortical layers 2/3. To this end, we monitored the activity of excitatory neurons and two major classes of inhibitory interneurons, PV and SOM, in anesthetized APP/PS1 mice and non-transgenic littermates (WT) using the genetically encoded calcium indicator GCaMP.

We show that SOM interneurons are hyperactive, while PV and excitatory neurons are hypoactive in APP/PS1 mice. Furthermore, SOM interneurons located near amyloid plaques are more active than those located farther away. Finally, pairwise neuronal correlations of excitatory neurons show a strong trend for a decrease in APP/PS1 mice.

## Results

### SOM interneurons are hyperactive near amyloid plaques in APP/PS1 mice.
The impact of AD on inhibitory interneuron cell-type-specific spontaneous activity is not well understood, with cortical SOM interneuron activity totally unexplored in vivo. Here, we performed in vivo multiphoton calcium imaging in anesthetized APP/PS1 (APP) mice and non-transgenic littermates (WT), to study cell-type-specific spontaneous neuronal activity.

We first monitored calcium transients in the somas of SOM interneurons using the genetically encoded calcium indicator jGCaMP7s (Fig. 1a). To target SOM interneurons selectively, we injected FLEX-jGCAMP7s into the somatosensory cortex of WT-SOM and APP-SOM mice (Fig. 1a). SOM interneurons exhibited spontaneous calcium transients in WT-SOM (Fig. 1b) and WT-APP (Fig. 1c) mice. We then estimated the number of deconvolved calcium events as a proxy measure for neuronal action potential firing (Supplementary Figure 1, Supplementary Movie 1). Due to the slower timescale of calcium indicator responses relative to action potential-related voltage changes, deconvolution of the calcium signal to the best estimate of spiking activity is required[33,34]. Several calcium deconvolution algorithms have been developed to address this non-trivial issue[36–39]. We chose to use a non-negative deconvolution algorithm[39,40] to estimate the timing and the number of deconvolved calcium events in our recordings because it outperformed all other algorithms in a dataset of mutual calcium imaging and ground-truth electrophysiology recordings[41]. Our results show that SOM interneuron event rates were 2.7-fold higher in APP-SOM mice relative to WT-SOM mice (Fig. 1d, e). The fraction of inactive SOM cells was not different between conditions (Fig. 1f). Thus, SOM interneurons are hyperactive in APP/PS1 mice.

Studies examining spontaneous activity in indiscriminate neuronal populations in vivo have shown that neurons near amyloid plaques at distances less than 60 μm are more active than those farther away[5]. Therefore, to understand the relationship between amyloid plaques and the observed hyperactivity, we measured the correlation between SOM interneuron event rates and distance to amyloid plaques in APP/PS1 mice. We found a significant negative correlation between SOM interneuron activity and distance to amyloid plaques in APP/PS1 mice (Fig. 1g). SOM interneurons near amyloid plaques (<60 μm) were more active compared to those located away from plaques (Fig. 1h). Therefore, SOM interneuron hyperactivity is correlated with proximity to amyloid plaques.

Postmortem immunohistochemical analysis of WT-SOM mice confirmed that ~90% of jGCaMP7s cells were positive for SOM (Supplementary Fig. 2a, b), thus jGCaMP7s targeted SOM interneurons with high selectivity.

### PV interneurons are hypoactive in APP/PS1 mice.
We then asked whether the spontaneous hyperactivity in the somatosensory cortex can be generalized to other interneuron subtypes in APP/PS1 mice in vivo. PV interneurons have markedly different electrophysiological properties relative to SOM interneurons as they can reach very high firing rates[42]. Hence, we next used similar experimental procedures (Fig. 2a) and targeted jGCaMP7s expression to PV interneurons in the somatosensory cortex of WT-PV-Cre (WT-PV) (Fig. 2b) and APP-PV-Cre (APP-PV) mice (Fig. 2c). Mean event rates were 5.6-fold higher in WT-PV interneurons (Fig. 2d, Supplementary Movie 2, and Supplementary Fig. 3) relative to WT-SOM interneurons (Fig. 1d; mean ± s.e.m.: 0.18 ± 0.03 for PV interneurons, and 0.032 ± 0.01 for SOM

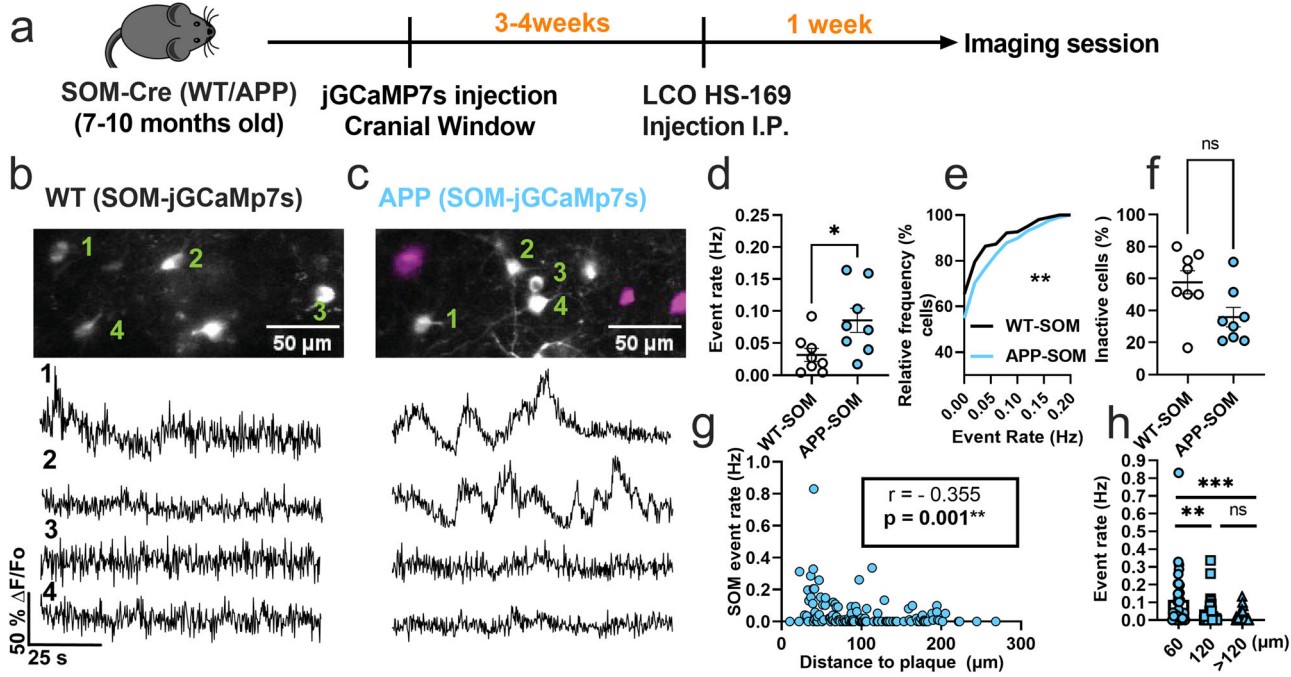

**Fig. 1 SOM interneurons are hyperactive in APP/PS1 mice relative to WT mice. a** Timeline of the experimental procedures. **b, c** Top, in vivo two-photon fluorescence images of jGCaMP7s (grey) in SOM expressing interneurons in layer 2/3 of the somatosensory cortex from WT (**b**) and APP/PS1 (**c**) mice. Amyloid plaques were labeled with LCO-HS-169 (magenta); Scale bars, 50 μm. Bottom, representative normalized fluorescence traces of SOM interneurons activity from control (**b**) and APP/PS1 (**c**) mice. Representative images were created by averaging 500 images from SOM interneuron recordings at 256×256 resolution with green numerical labels to the right of each representative cell. **d** Mean neuronal activity rates as determined by counting the rate of deconvolved $Ca^{+2}$ events (Mann–Whitney $U = 11$, $p = 0.024$, two-tailed, $n = 8$ WT-SOM mice; 8 APP-SOM mice). **e** Cumulative frequency distribution of event rates in all imaged neurons (KS-test:D = 0.152, $p = 0.0044$, $n = 210$ WT-SOM neurons; 364 APP-SOM neurons). **f** Quantification of the fraction of inactive cells in APP-SOM mice relative to WT_SOM (Mann–Whitney $U = 14$, $p = 0.06$, two-tailed, $n = 8$ WT-SOM mice; 8 APP-SOM mice). **g** Correlations between event rates and the distance to amyloid plaque center in SOM interneurons (Spearman correlation: $r_{(124)} = −0.355$, $p < 0.0001$. **h** Quantification of pooled event rates as a function of distance to amyloid plaque. Neurons were pooled into three groups depending on their distance to amyloid plaque: <60 μm, 60–120 μm, and >120 μm (Kruskal–Wallis test, $H_{(2)} = 18.41$, $p < 0.0001$). Post hoc multiple comparisons were $p = 0.0028$ for SOM event rates <60 μm vs. SOM < 120 μm and $p = 0.0001$ for SOM event rates <60 μm vs. rates >120 μm and not significant, $p = 0.715$, for SOM event rates <120 μm vs. rates >120 μm. Each solid circle in **d**, **f** represents an individual animal, while each circle in **g**, **h** represents an individual neuron. All error bars reflect the mean ± s.e.m. Asterisks denote statistically significant differences (**$p < 0.01$, ****$p < 0.0001$), while 'ns' denotes no significance $p > 0.05$.

interneurons). Contrary to our findings in SOM interneurons, PV interneurons showed a 1.8-fold decrease in event rates in APP-PV mice relative to WT-PV mice (Figs. 2d, e). The fraction of inactive PV cells was comparable across conditions (Fig. 2f). PV interneuron activity did not correlate with proximity to amyloid plaques (Fig. 2g, h). Therefore, PV interneurons are hypoactive in APP/PS1 mice irrespective of their proximity to amyloid plaques. These results suggest a cell-type-specific disruption in cortical interneuron activity of APP/PS1 mice.

Postmortem immunohistochemical analysis of PV-Cre mice confirmed that 86% of jGCaMP7s cells were positive for PV (Supplementary Fig. 2c, d).

**Excitatory neurons are hypoactive in APP/PS1 mice.** The disrupted interneuronal activity of APP/PS1 mice prompted us to investigate whether this deficit is related to E/I imbalance. SOM and PV interneurons form inhibitory synaptic contacts onto excitatory neurons, therefore dominant hypoactivity in either or both interneuron subtypes can potentially result in hyperactivity of excitatory cells. Because of the numerous reports of hyperactive cells in APP/PS1 mice in vivo, we hypothesized that excitatory cells are hyperactive as a result of dominant hypoactivity in PV interneurons. To test this hypothesis, we imaged spontaneous activity of layer 2/3 excitatory neurons of the somatosensory cortex in WT (WT-EX) and APP/PS1 (APP-EX) mice. We targeted excitatory neurons using a human CaMKIIα promoter to drive the expression of GCaMP6s (Fig. 3a–c). These recordings were performed on a combination of SOM-and PV-Cre mice to ensure comparability to interneuron recordings (see methods). In contrast to our hypothesis, excitatory neurons of APP-EX mice showed 1.6-fold lower event rates (Fig. 3d and Supplementary Movie 3) relative to WT-EX mice, with an overall lower cumulative distribution of event rates across all recorded cells (Fig. 3e). Mean event rates of excitatory cells were stable over 2 to 3 months (Supplementary Fig. 4) and did not depend on sex or mouse model (Supplementary Fig. 5). No significant change in the fractions of inactive excitatory cells (Fig. 3f) was observed. Excitatory neuron activity did not depend on amyloid plaque distance (Fig. 3g, h and Supplementary Movie 4). Therefore, cortical excitatory neurons are hypoactive in APP/PS1 mice irrespective of their proximity to amyloid plaques. To confirm that the cell-type-specific findings reported here are not driven by the analysis method, we reanalyzed the normalized fluorescence traces for all cell types using peak counting[5,43]. Peak counting showed similar results compared to non-negative deconvolution for excitatory and SOM interneurons (Supplementary Fig. 6). Although PV interneuron activity was not significantly different between groups using peak counting (Supplementary Fig. 6c), the cumulative frequency distribution of event rates was different in PV-APP interneurons relative to PV-WT interneurons

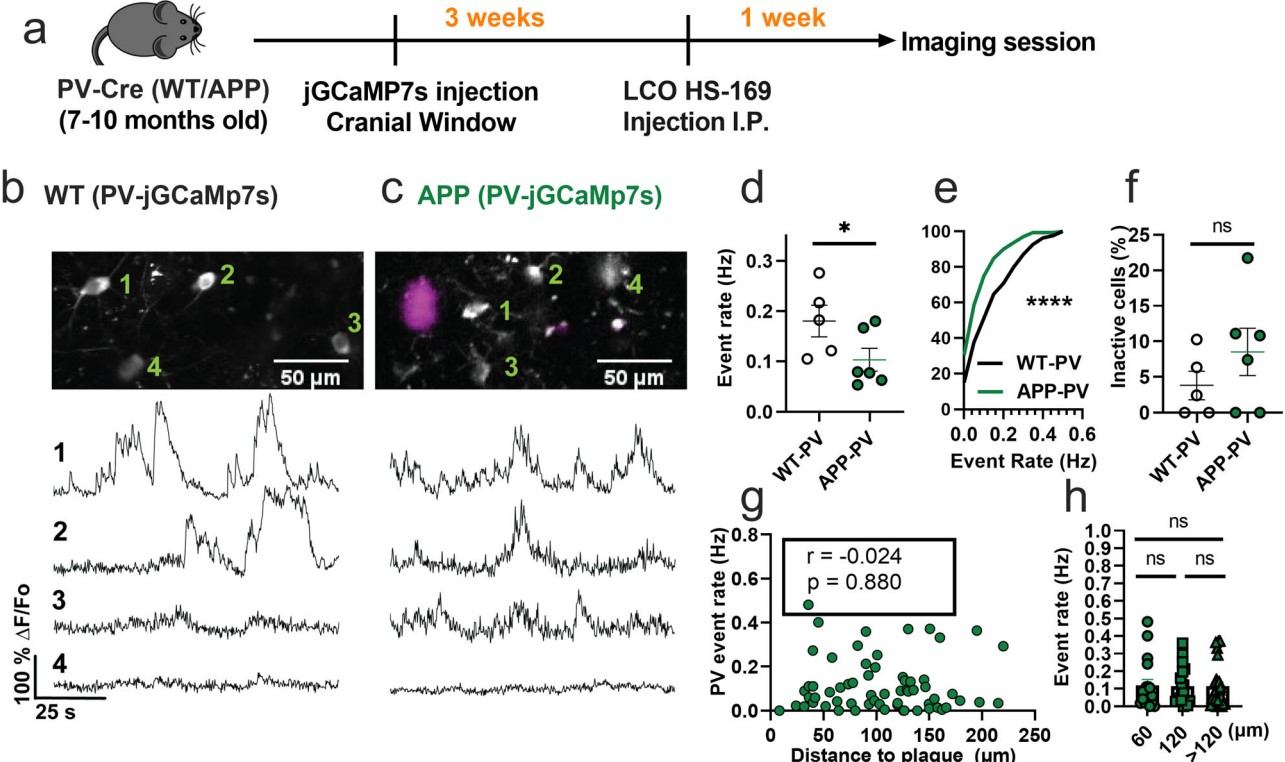

**Fig. 2 PV interneurons are hypoactive in APP/PS1 mice relative to WT mice. a** Timeline of the experimental procedures. **b**, **c** Top, in vivo two-photon fluorescence images of jGCaMP7s in PV expressing interneurons (grey) in layer 2/3 of the somatosensory cortex from WT (**b**) and APP/PS1 (**c**) mice. Amyloid plaques were labeled with LCO- HS-169 (magenta); Scale bars, 50 μm. Bottom, representative normalized fluorescence traces from control (**b**) and APP/PS1 (**c**) mice. Representative images were created by averaging 500 images from PV interneuron recordings at 256×256 resolution with green numerical labels to the right of each representative cell. **d** Mean event rates (Mann–Whitney $U = 4$, $p = 0.051$, two-tailed, $n = 5$ WT-PV mice; 6 APP-PV mice). **e** Cumulative frequency distribution of event rates in all imaged neurons (KS-test:$D = 0.268$, $p < 0.0001$, $n = 254$ WT-PV neurons; 172 APP-PV neurons). **f** Quantification of the fraction of inactive cells (Mann–Whitney $U = 4$, $p = 0.28$, two-tailed) in imaged WT-PV and APP-PV ($n = 5$ WT mice; 6 APP/PS1 mice). **g** Correlations between event rates and the distance to amyloid plaque center in PV interneurons (Spearman correlation: $r_{(63)} = −0.024$, $p = 0.88$), and **h** quantification of pooled PV event rates as a function of amyloid plaque distance (Kruskal–Wallis test, $H_{(2)} = 0.46$, $p < 0.79$). Each solid circle in **d**–**f** represents an individual animal, while each circle in **g**, **h** represents an individual neuron. All error bars reflect the mean ± s.e.m. Asterisks denote statistically significant differences (*$p < 0.05$, ****$p < 0.0001$), while 'ns' denotes no significance $p > 0.05$.

(Supplementary Fig. 6d). In summary, while the analysis method has a minimal effect on most of our findings, PV interneuron results are limited and should be interpreted cautiously.

**Pairwise neuronal correlations in APP/PS1 mice.** To evaluate the impact of amyloid accumulation on the activity synchrony of different cell types, we calculated the Pearson correlation values (ρ) for all possible pairwise combinations of active neurons within the somatosensory cortex. Mean correlation values showed a trend for a decrease in APP-EX relative to WT-EX mice (Fig. 4a and Supplementary Figs. 7a and 8). It is established that neuronal synchrony decreases as the distance between neurons increases[44]. Therefore, we determined the relationship between correlation values and pairwise distance between neurons (Fig. 4b). As expected, correlation values decreased as a function of increased inter-neuronal distance, a phenomenon that seemed to be similar in WT-EX and APP-EX mice (Fig. 4b and Supplementary Fig. 7b). Since correlations are known to increase when neuronal event rates increase[45], we also examined the relationship between mean correlations of a given neuron and its event rate. Excitatory neuronal ρ values increased as function of neuronal event rates (Fig. 4c, and Supplementary Fig, 7c). These observations did not significantly differ in WT-EX and APP-EX mice (Fig. 4c and Supplementary Fig. 7c), suggesting that the decreased correlation values in APP-EX mice are related to decreased event rates and

not synchrony. We calculated significantly fewer pairwise correlations for interneurons relative to excitatory cells because only a few cells could be recorded in each imaging frame. Nevertheless, there was no significant change in correlation values for SOM and PV interneurons in APP/PS1 mice relative to WT (Fig. 4d, e and Supplementary Fig. 7d, e).

## Discussion

Altered neuronal network activity in AD has been observed in several cerebral amyloidosis mouse models and AD patients[5,6,11,12,20,29,46,47]. Recent reports suggest that inhibitory circuits are particularly vulnerable, resulting in E/I imbalance and circuit-level dysfunction in AD[19,29,48,49]. However, the exact nature of the disrupted inhibitory activity is not yet fully understood in vivo. We report cell-type-specific interneuron dysfunction during spontaneous activity recorded in vivo in APP/PS1 mice. SOM interneurons were hyperactive while PV interneurons were hypoactive. Additionally, the changes in SOM interneuron activity, but not PV activity, correlated with proximity to amyloid plaques. These inhibitory deficits were accompanied by a decrease in excitatory neuronal activity, thus providing a detailed picture of the nature of E/I imbalance in the APP/PS1 mouse model.

Our finding of decreased excitatory neuronal activity in APP/PS1 mice may appear to conflict with reports of hyperactivity in

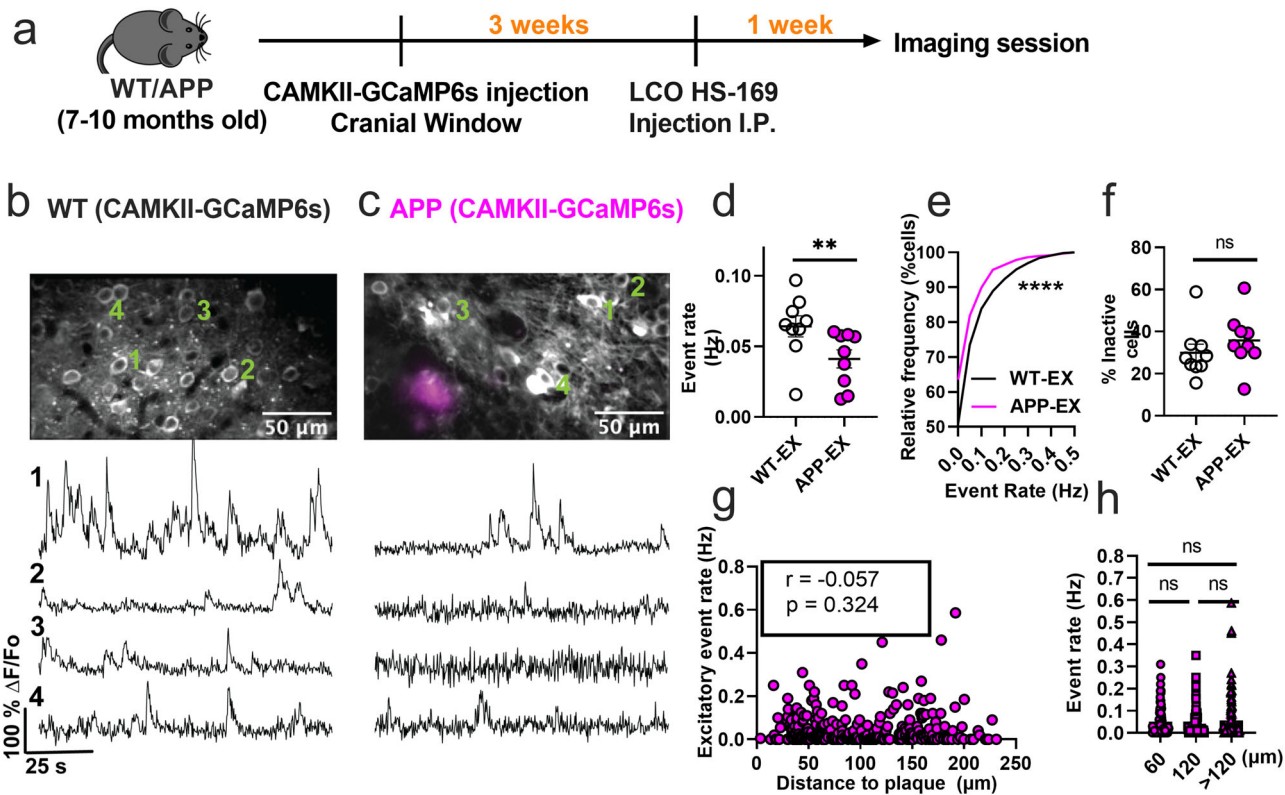

**Fig. 3 Excitatory neurons are hypoactive in APP/PS1 mice. a** Timeline of the experimental procedures. **b**, **c** Top, in vivo two-photon fluorescence images of GCaMP6s in CaMKII expressing (excitatory) neurons (grey) in layer 2/3 of the somatosensory cortex from non-transgenic controls (**b**) and APP/PS1 (**c**) mice. Amyloid plaques were labeled with LCO- HS-169 (magenta); Scale bars, 50 μm. Bottom, representative normalized fluorescence traces from control (**b**) and APP/PS1 (**c**) mice. Representative images were created by averaging 500 images from excitatory neuron recordings at 256×256 resolution with green numerical labels to the right of each representative cell. **d** Mean neuronal activity rates as determined by counting the rate of deconvolved $Ca^{+2}$ events (Mann–Whitney $U = 11$, $p = 0.0078$, two-tailed, $n = 9$ WT-EX mice; 9 APP-EX mice). **e** Cumulative frequency distribution of event rates in all imaged neurons (KS-test:D = 0.129, $p < 0.0001$, $n = 1412$ WT-EX neurons; 1603 neuron APP-EX neurons). **f** Quantification of the fraction of inactive cells (Mann–Whitney $U = 23$, $p = 0.136$, two-tailed) in imaged WT-EX and APP-EX ($n = 9$ WT-EX mice; 9 APP-EX mice). **g** Correlations between event rates and the distance to amyloid plaque center in excitatory neurons (Spearman correlation: $r_{(298)} = -0.057$, $p = 0.324$), and **h** quantification of pooled excitatory neurons event rates as a function of amyloid plaque distance (Kruskal-Wallis test, $H_{(2)} = 0.46$, $p < 0.79$). Each solid circle in **d**–**f** represents an individual animal, while each circle in **g**–**h** represents an individual neuron. All error bars reflect the mean ± s.e.m. Asterisks denote statistically significant differences (*$p < 0.05$, ****$p < 0.0001$), while 'ns' denotes no significance $p > 0.05$.

amyloidosis models[5,20,50]. This discrepancy could be related to differences in animal model strains. Amyloidosis models such as APP/PS1 mice on C57BL/6J background[20] exhibit seizures[51]. Genetic background is known to influence susceptibility to seizures in amyloidosis models[52], and it is unclear if the seizure phenotype reported in C57BL/6J APP/PS1 mice would also be observed in APP/PS1 mice on C57BL/6;C3H background used in the current study[53]. Since seizures could be accompanied by increased excitability of pyramidal cells[51,54], higher seizure incidence could result in a dominant hyperactivity phenotype in an amyloidosis mouse model. Furthermore, in agreement with our PV and excitatory cells findings, the initial reports of the hyperactivity phenotype also reported an increased fraction of hypoactive cells[5,55]. On the other hand, the decrease in spontaneous activity of PV interneurons reported in this study using the deconvolution method is in agreement with previous reports linking PV interneuron deficits to gamma rhythm disruption in AD[6,27,29,30,56]. It is important to note that our PV findings are limited and uncertain since the peak counting method did not show a significant effect. Future studies should measure PV interneuron firing rates in AD models using in vivo electrophysiological recordings to validate our findings.

Mounting evidence from in vivo and in vitro studies has shown that circuit activity is homeostatically regulated to maintain

neuronal firing rates constrained to a particular functional limit[57,58]. Hence, neuronal circuits will respond to changes in firing activity, particularly in excitatory cells, with compensatory mechanisms to restore the perturbed firing homeostasis[57,58]. The process of restoring firing homeostasis involves two well-documented mechanisms. One, the circuit can adjust synaptic strength by gradually changing the ratio of inhibitory and excitatory synapses in the direction that normalizes the perturbation, or two, it can modify the intrinsic excitability to balance synaptic input and firing rates[57].

We provide evidence for the failure of homeostatic mechanisms to maintain appropriate neuronal firing in APP/PS1 mice. Because the hyperactivity of SOM interneurons correlates with proximity to amyloid plaques SOM interneurons can potentially drive this dyshomeostasis since they supply aberrant inhibition to layer 2/3 pyramidal cells[26,59,60]. SOM interneurons were also reported to strongly inhibit PV interneurons in cortical layers 2/3 during visual processing[61]. SOM interneuron hyperactivity could thus suppress PV interneuron firing in APP/PS1 mice. Indeed, we saw hypoactivity within PV interneurons.

In addition to the synaptic mechanisms, alterations in the intrinsic excitability of different cell types can account for the changes in their activity patterns[57]. While the evidence for altered intrinsic excitability of excitatory cells in amyloidosis mouse

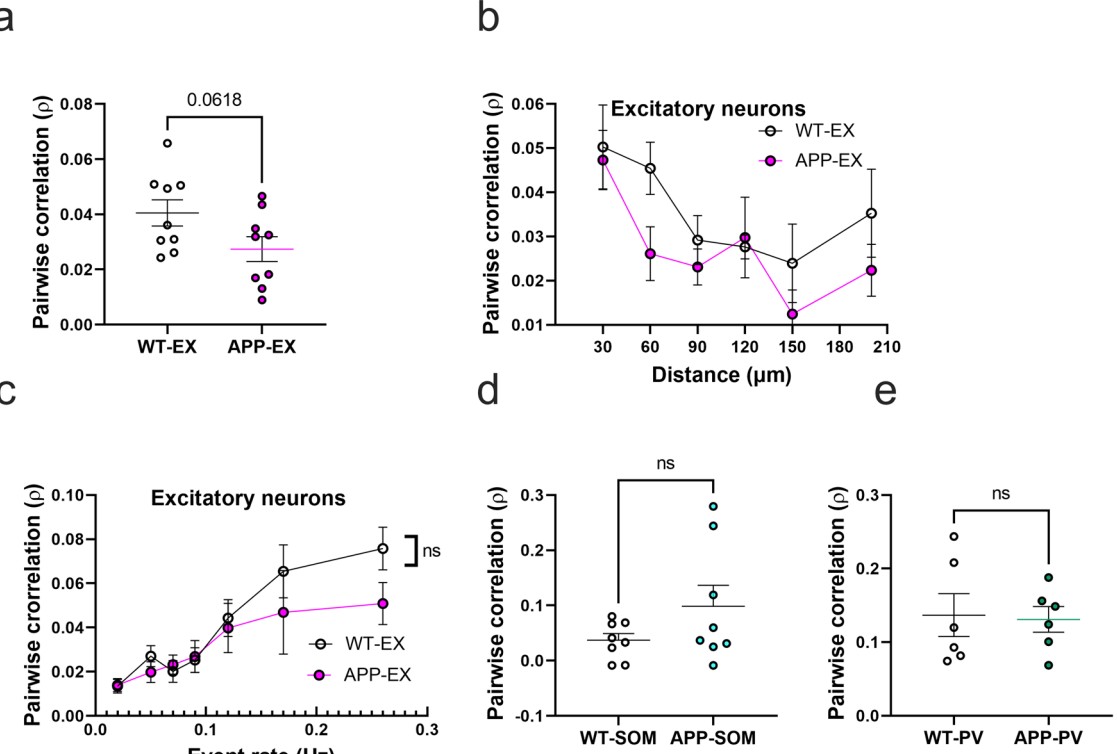

**Fig. 4 Pairwise neuronal correlations in APP/PS1 mice. a** Average pairwise Pearson correlation values for excitatory neurons in APP/PS1 and WT mice ($t_{(16)} = 2.008$, $n = 9$ WT-EX mice, $n = 9$ APP-EX mice, $p < 0.062$, two-tailed). **b** Relationship between pooled pairwise Pearson correlation values and pairwise interneuronal distance (Two-Way ANOVA, main distance effect $F_{(5, 87)} = 4.5$, $p = <0.001$, main genotype effect $F_{(1, 87)} = 4.38$, $p = 0.039$, main genotype and distance interaction $F_{(5, 87)} = 0.62$, $p = 0.682$). The x axis values represent the maximum limit of the pooled pairwise distances. **c** Relationship between correlation values and event rates after pooling data according to geometric means of neuronal event rates (two-way ANOVA, main event rate effect $F_{(6,107)} = 9.7$, $p < 0.0001$, main genotype effect $F_{(1, 107)} = 24$, $p = 0.1223$, main genotype and event rate interaction $F_{(6, 107)} = 0.7$, $p = 0.58$). **d, e** Average pairwise Pearson correlation values for SOM interneurons (**d**) ($t_{(14)} = 1.54$, $n = 8$ WT-SOM mice, $n = 8$ APP-SOM mice, $p = 0.147$, two-tailed) and PV interneurons (**e**) ($t_{(10)} = 0.16$, $n = 6$ WT-PV mice, $n = 6$ APP-PV mice, $p = 0.95$, two-tailed) in APP/PS1 and WT mice. All error bars reflect the mean ± s.e.m.

models has been inconsistent[19,62,63], a decrease in intrinsic excitability of pyramidal cells could potentially account for their hypoactivity in APP/PS1 mice.

Our results have important implications for future studies targeting the inhibitory circuit as a treatment strategy for AD. While inhibitory circuit deficits are prevalent in AD, it is not clear whether a therapeutic strategy that broadly restores inhibitory tone, using GABA agonists or antagonists, would benefit early-stage to intermediate-stage AD patients. Our results suggest that a more targeted approach to restoring inhibitory tone will be needed to overcome these network deficits since interneuron cell types were differentially affected by AD pathology. Future studies using approaches to control neuronal activity, such as optogenetics, should aim to decrease SOM or increase PV interneurons activity. Indeed, several studies have targeted PV interneurons activation using optogenetic and pharmacological approaches to restore oscillatory gamma activity and improve cognitive functions in preclinical animal models[27,30,56,64].

Our study assumes that the $Ca^{+2}$ indicator (GCaMP) concentration distribution across cells in different models is similar. A violation of this assumption could result in inaccurate estimation of event rates due to changes in indicator sensitivity and potentially cellular $Ca^{+2}$ buffering capacity[65]. We opted to perform our experiments under low isoflurane anesthesia to reduce experimental variability[66] and compare our findings to previous reports of altered single-cell activity in AD[5,20]. However, anesthesia is known to reduce neuronal activity and presents a

limitation to our current findings[44]. Therefore, future experiments should include reproducing these findings in awake behaving animals using calcium imaging and multi-electrode array electrophysiology.

Together, these findings show an association between amyloid pathology and interneuron-related circuit deficits in an Alzheimer's disease mouse model. These results should guide future therapeutic approaches targeting circuit dysfunction at the early to intermediate stages of Alzheimer's disease.

## Methods

**Animals**. All experimental procedures were approved by MGH Institutional Animal Care and Use Committee (IACUC). We crossed homozygote APPswe/PSdE9 transgenic mice overexpressing the amyloid precursor protein with the Swedish mutation and deltaE9 mutation in presenilin 1 (APP/PS1) (Stock #034829, C57BL/6;C3H genetic background, The Jackson Laboratory)[67] with homozygote SOM-IRES-Cre knock-in mice (Stock #013044, The Jackson Laboratory)[68] or homozygote PV-Cre mice (Stock # 008069, The Jackson Laboratory)[69] to generate APP-SOM and APP-PV mice, respectively. Age-matched non-transgenic littermates were used as controls (WT-SOM and WT-PV). Experiments were performed in 8–11-month-old animals including males and females. Prior to any surgical procedures, animals of the same sex were housed in up to 4 animals/cage. Individual housing was maintained after surgical manipulation. All animals had ad libitum access to water and food and were maintained on a 12/12-hour day/night cycle in a pathogen-free environment.

**Cranial window surgery and viral transduction of genetically encoded Ca indicator**. Mice were initially anesthetized with 5% isoflurane inhalation in O2 and maintained on 1.5% isoflurane during the surgery. Throughout the procedure, mice

were placed on a heating pad to keep the body temperature at ~37.5 °C and their eyes were protected with an ophthalmic ointment. A 5 mm craniotomy was drilled over the right somatosensory cortex using sterile technique. The dura was kept intact and wet with saline.

To express the ultrasensitive genetically encoded fluorescent calcium indicators in cortical inhibitory interneurons, we injected FLEX-jGCaMP7s (Addgene# 104491-AAV1)[39] in SOM-CRE ($4 \times 10^{12}$ genome copies/ml) and PV-CRE ($1 \times 10^{12}$ genome copies/ml) WT and APP mice ($n = 5$–8 per group) at ~300 μm below the surface in the somatosensory cortex (2 sites, at 1.5- and 2.5-mm posterior to bregma and 2 mm lateral to the midline suture). We then used adeno-associated virus AAV1 carrying the construct GCaMP6s under the CaMKIIα promoter (Addgene# 107790-AAV9, $1 \times 10^{12}$ genome copies/ml)[40] to target excitatory cells of WT and APP mice ($n = 9$ per group). To ensure our excitatory neurons experiments are comparable to our interneurons experiments, CaMKIIα-GCaMP6s injections were targeted to a combination of PV-CRE ($n = 5$ per group) and SOM-CRE ($n = 4$ per group) WT and APP mice All the injections were delivered using a Hamilton syringe and Pump 11 Elite microsyringe pump (Harvard Apparatus) at a rate of 200 nl/min. Each injection consisted of ~1 μl of viral construct diluted in phosphate-buffered saline containing 0.01% Pluronic F-68. After injecting the desired volume, the Hamilton syringe was left at the injection site for 5 min to prevent backflow of the viral solution. A round glass coverslip (5 mm diameter) was placed over the right somatosensory cortex using a mix of dental cement and cyanoacrylate. Following each surgery, mice were housed individually and received analgesia (buprenorphine and acetaminophen) for 3 days postoperatively.

**In vivo multiphoton calcium imaging**. Imaging was performed under light iso-flurane anesthesia to reduce experimental variability[12]. Mice were initially sedated with 5% isoflurane in room air using the SomnoSuite®Low-Flow Anesthesia System (Kent Scientific). Mice were then imaged under light anesthesia and low air-flow rates (1% isoflurane and ~40 mL/min air flow for a 30 g mouse). A heating pad was used to keep the animal's body temperature at 37.5 °C, and ophthalmic ointment was used to protect the eyes. Animals were kept for at least 60 min under light isoflurane anesthesia before imaging. Mice typically showed an absence of tail-pinch reflex and a respiration rate of 80–120 breaths per minute during the imaging session.

A Fluoview FV1000MPE multiphoton microscope (Olympus) with a mode-locked MaiTai Ti:sapphire laser (Spectra-Physics) set to 900 nm was used for two-photon imaging. Spontaneous $Ca^{2+}$ fluorescence signals from the somatosensory cortex were collected at 5 Hz through a 25x, 1.05 numerical aperture water immersion objective (Olympus) at 2x digital zoom. The Fluoview program was used to control the scanning and image acquisition (Olympus). Multiple fields of view ($\approx 160 \times 100$ μm, 1 pixel per μm) were imaged per mouse, and each field of view was recorded for at least 100 s.

**Calcium imaging data analysis**. We used Suite2p[70] and a custom-written MATLAB program to analyze spontaneous calcium events as follows. Recordings were collected as Olympus Image Format (OIF) files, converted to Tiff files using ImageJ, then imported to Suite2P for automated registration, somatic regions of interest (ROI) detection, and neuropil contamination estimation. We eliminated the neuropil contribution to emphasize neuronal soma signals. All images were aligned using the Suite2p rigid registration function that computes the shifts between each frame and a reference image using the phase correlation method. ROIs were automatically detected from the mean image for each recording using Suite2p anatomic detection methods. All automated ROIs were inspected manually, and missing ROIs were added. Fluorescence values were extracted from each ROI for each frame, and the mean for each cell was computed. In addition, neuropil contamination was estimated from at least 100 pixels surrounding each somatic ROI using Suite2p. This gave us two vectors of fluorescence values for the soma and the neuropil. Data were then analyzed using a custom-written MATLAB program (https://github.com/moustaam0/Algamal2022_analysis_w_OASIS) that was adopted from prior calcium imaging studies[35] as described in the following paragraphs.

The neuropil vector was weighed a factor of 0.7[34]. The weighted neuropil vector was subtracted from the somatic vector to produce a corrected vector of fluorescence values. The normalized ΔF/F trace was calculated according to the equation $(Fi - F_0)/F_0)$, where $Fi$ is the frame index, and $F_0$ is the fluorescence baseline. To measure baseline $F_0$ fluorescence in each cell, we employed a moving average baselining function like prior calcium imaging investigations[35]. The baselining function applies a sliding window of 250 frames (50 s) and quartile cut-off values ranging from the bottom tenth percentile to the median, depending on how active the neuron was. We then counted the total number of deconvolved events from the generated ΔF/F traces to estimate neuronal event rates using a robust non-negative deconvolution approach[40] (https://github.com/zhoupc/OASIS_MATLAB). The deconvolution was employed using a FOOPSI Autoregressive model#1 (AR1) with the event size constrained to be 2.5 times the noise levels. GCaMP6/7s decay time was set at 1.25 s to estimate the 'γ' parameter, while sparsity penalty parameter ($\lambda$) was set to 0.

To calculate the correlation coefficients, deconvolved calcium events from each neuron were binarized and corrected for pixel timing using linear interpolation[71]. Correlation coefficients were then computed as the Pearson correlation coefficient

($\rho$) for all possible pairings of active neurons (event rate higher than 0.01 Hz) using MATLAB 'corr' function. All possible $\rho$ values were pooled per experimental group to estimate the degree of synchrony. Distances between individual neurons represent the Euclidean distance between the ROI centroids. To estimate the effect of distance to each amyloid plaque on neuronal activity, we chose fields of view containing several neuronal cells and a single amyloid plaque in the same plane. Additionally, we made sure that no additional amyloid plaques were located within 100 μm in the x,y,z planes of the imaging field of view. Distances were then calculated as the Euclidean distances between the amyloid plaque center and the neuronal ROI centroid.

**Immunohistochemistry**. After the acquisition of multiphoton images, SOM-Cre and PV-Cre mice injected with FLEX-jGCaMP7s were euthanized using $CO_2$ inhalation. Animals were then transcardially perfused, their brains collected and kept in 4% paraformaldehyde (PFA)/ phosphate buffered saline (PBS) solution overnight. The following day, brains were washed in PBS then transferred to a cryopreserving solution (30% sucrose in PBS). 40 μm thick coronal slices of mouse brains were subjected to antigen retrieval in citrate buffer. The sections were subsequently permeabilized with Triton X-100, blocked with normal goat serum (NGS), and incubated overnight at 4 °C with the following primary antibodies; PV (mouse monoclonal anti-PV, 1:500; Sigma P3088), SOM (rat polyclonal anti-SOM, 1:50; MAB354), or GFP (chicken polyclonal anti-GFP, 1:500, A10262). Tissue sections were then washed and incubated with their respective secondary antibodies (1:500) for 1 hour at room temperature. The slides were then mounted with Vectashield antifade mounting medium (Vector Laboratories). A Fluoview FV3000 confocal microscope (Olympus) with a ×40 objective was used to image PV and SOM-stained sections. Quantification of SOM and PV positive cells was done manually.

**Statistics and reproducibility**. The appropriate statistical test was chosen after the assessment of normal distribution using the Shapiro-Wilk normality test. In most cases, a non-parametric test was used because normal distribution could not be assumed. For comparison involving two experimental groups, we used either a parametric two-sided Student's $t$ test or a non-parametric Mann–Whitney $t$-test as detailed in the figure captions. Comparisons between three groups were assessed using Kruskal–Wallis one-way analysis of variance followed by a Dunn's test to correct for multiple comparisons. Two-Way ANOVA was used to assess the relationship between inter-neuronal distances and pairwise neuronal activity correlations. Spearman correlation was used to assess the relationship between amyloid plaque distance and event rates. A p value of less than 0.05 was considered statistically significant. Individual animals were used to preform statistical analysis on all data presented in the main figures (5–9 mice per group).

**Reporting summary**. Further information on research design is available in the Nature Portfolio Reporting Summary linked to this article.

## Data availability

All data needed to drive the conclusions of this article are present in the paper and the Supplementary Information. Additionally, numerical source data for main figures and all unprocessed calcium traces (MATLAB source files) are available[72,73] at https://figshare.com/account/home#/collections/6290574.

## Code availability

The MATLAB code used to generate data presented in this article available at (https://github.com/moustaam0/Algamal2022_analysis_w_OASIS).

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

## Acknowledgements

This work was funded by NIA (RF1AG061774). Acknowledgement is made to the donors of the Alzheimer's Disease Research Program, a program of the BrightFocus Foundation, for support of this research under grant A2021001F and Alzheimer's Association under grant AARG-18-52336.

## Author contributions

Conception and design of study (M.A., K.V.K., and B.J.B.), mouse surgery (M.A. and A.N.R.), in vivo imaging and maintenance of two-photon setup (M.A. and S.S.H.), data analysis (M.A. with help from M.M., L.M., and Q.Z.), coding and data interpretation (M.A.), immunohistochemistry (M.R.M.), manuscript preparation (M.A. with help from A.N.R., B.J.B., and K.V.K.), securing funding (K.V.K., B.J.B., and D.G.) and project supervision (B.J.B. and K.V.K.).

## Competing interests

The authors declare no competing interests.
