## [Peer Review File · Communications Biology]

Reviewers' comments:

Reviewer #1 (Remarks to the Author):

In this manuscript, the authors used cross-breeding of the APP/PS1 mouse model of Alzheimer's Disease with interneuron-specific lines to examine spontaneous activity changes of excitatory and PV and SST inhibitory neuron classes under isoflurane anesthesia. They conclude that SST interneurons are 'hyperactive', whereas pyramidal neurons and PV interneurons are 'hypoactive'.

Results are reported exclusively from deconvolved calcium transients. This is not convincing. They first need to perform simultaneous patch recordings from the GCaMP expressing excitatory and inhibitory neurons for a ground truth calibration of their calcium signals that then can inform the deconvolution to estimate the underlying spike rates. The lack of ground truth data is a significant problem of the manuscript because all conclusions rely on this approach (which is also not well described and not validated). Another major limitation is that even if the results hold, there is a lack of mechanistic insight; mainly, how the cell type specific changes are causally linked must be worked out before publication.

It is also relevant for the paper that, particularly for interneurons, the brain state matters a lot, and the anesthesia is a particularly strong confound when studying them. This can be addressed by performing awake recordings.

Figure 1 b/c, top panel - please clarify how these images were generated (for all the subsequent in vivo images). There is almost no background, and GCaMP-expressing neurons are saturated and have a filled nucleus which can be a sign of toxicity.

Figure 1 b/c, bottom panel - please clarify why the example neurons have different noise levels although they are from the same recording? This is also an issue in several of the subsequent example traces.

Figure 4 - Please clarify how this analysis was done and show data for the biological replicates (i.e., animals). The strong significance is because neurons from all animals were pooled in the analysis. Does it appear that much of the effect is driven by an increase in pairwise correlations at 210 μ m in WT (see panel b)? This is puzzling to me as one would not expect an increase in correlation with distance?

"We propose a mechanism by which amyloid plaque-driven hyperactivity in SOM interneurons leads to decreased firing of pyramidal cells." This is a misleading statement because no mechanistic link has been established.

"It is important to note that the hyperactivity phenotype has been observed in amyloidosis mouse models exhibiting a considerable number of seizures, and it remains unknown whether this phenotype occurs in amyloidosis models lacking the seizure phenotype." The statement suggests that this study addresses this issue; however, they use APP/PS1 mice known to have seizures. Indeed, they refer to a JAX datasheet that the APP/PS1 line on this particular background does not have seizures; however, I am not convinced they can rule out the presence of seizures in their cohort. They need to demonstrate the absence of seizures electrographically in their cohort if they wish to make such claims (it is, for example, known that animal housing affects the incidence of seizures).

"It is possible that most of the previously reported hyperactive cells were, in fact, SOM interneurons ." This is unlikely since the number of SST neurons is much smaller than the number of hyperactive neurons reported in the previous literature, as far as I can ascertain.

Reviewer #2 (Remarks to the Author):

Summary:

In the manuscript 'Hyperactive somatostatin interneurons near amyloid plaque and cell- type-

specific firing deficits in a mouse model of Alzheimer's disease', Algamal et al. explore changes in the activity of distinct inhibitory and excitatory neuron types in somatosensory cortex layer 2/3, in the well characterized APP/PS1 AD mouse model. Cell-type-specific differences and vulnerabilities in AD are becoming more apparent and appreciated as part of the disease process, thus this manuscript explores an important and timely topic which will be of wide interest to the field. Data from excitatory cells and two distinct interneuron types, as well as their spatial relationships to amyloid plaque deposits, makes for interesting comparisons. Generally, the manuscript is well organized, the methods are sound, and data is compelling. However several issues, mostly regarding interpretations and contextualization, should be addressed:

Major points:

-In multiple instances in the abstract and introduction, the authors state that early studies did not discriminate between neuronal subtypes (e.g.; 'the impact of amyloid pathology on distinct neuronal types remains unexplored within intact neuronal circuits'; 'several studies have assessed the firing activity of distinct neuronal cell types in amyloidosis mouse models'.) This is not fully accurate- several groups have explored physiological changes in several different AD models explicitly in distinct neuronal types (e.g., ex vivo in Petrache et al., 2019, Hijazi et al., 2019, Park et al., 2020, Verret et al., 2012, Olah et al. 2022 and in vivo, e.g., Bai et al. 2017, Martinez-Losa et al. 2018, Hanson et al. 2020). Throughout the results and manuscript in general, the authors should better frame the novelty of their work in the context of these studies (full refs here):

Bai, Y., Li, M., Zhou, Y., Ma, L., Qiao, Q., Hu, W., Li, W., Wills, Z.P., and Gan, W.-B. (2017). Abnormal dendritic calcium activity and synaptic depotentiation occur early in a mouse model of Alzheimer's disease. *Mol Neurodegeneration* 12, 1–15. <https://doi.org/10.1186/s13024-017-0228-2>.

Hanson, J.E., Ma, K., Elstrott, J., Weber, M., Sallet, S., Khan, A.S., Simms, J., Liu, B., Kim, T.A., Yu, G.-Q., et al. (2020). GluN2A NMDA Receptor Enhancement Improves Brain Oscillations, Synchrony, and Cognitive Functions in Dravet Syndrome and Alzheimer's Disease Models. *Cell Reports* 30, 381-396.e4. <https://doi.org/10.1016/j.celrep.2019.12.030>.

Hijazi, S., Heistek, T.S., Scheltens, P., Neumann, U., Shimshek, D.R., Mansvelder, H.D., Smit, A.B., and Kesteren, R.E. van (2019). Early restoration of parvalbumin interneuron activity prevents memory loss and network hyperexcitability in a mouse model of Alzheimer's disease. *Mol Psychiatry* 1–19. <https://doi.org/10.1038/s41380-019-0483-4>.

Martinez-Losa, M., Tracy, T.E., Ma, K., Verret, L., Clemente-Perez, A., Khan, A.S., Cobos, I., Ho, K., Gan, L., Mucke, L., et al. (2018). Nav1.1-Overexpressing Interneuron Transplants Restore Brain Rhythms and Cognition in a Mouse Model of Alzheimer's Disease. *Neuron* 0. <https://doi.org/10.1016/j.neuron.2018.02.029>.

Park, K., Lee, J., Jang, H.J., Richards, B.A., Kohl, M.M., and Kwag, J. (2020). Optogenetic activation of parvalbumin and somatostatin interneurons selectively restores theta-nested gamma oscillations and oscillation-induced spike timing-dependent long-term potentiation impaired by amyloid β oligomers. *BMC Biol* 18, 1–20. <https://doi.org/10.1186/s12915-019-0732-7>.

Petrache, A.L., Rajulawalla, A., Shi, A., Wetzell, A., Saito, T., Saido, T.C., Harvey, K., and Ali, A.B. (2019). Aberrant Excitatory–Inhibitory Synaptic Mechanisms in Entorhinal Cortex Microcircuits During the Pathogenesis of Alzheimer's Disease. *Cereb Cortex* 29, 1834–1850. <https://doi.org/10.1093/cercor/bhz016>.

Verret, L., Mann, E.O., Hang, G.B., Barth, A.M.I., Cobos, I., Ho, K., Devidze, N., Masliah, E., Kreitzer, A.C., Mody, I., et al. (2012). Inhibitory Interneuron Deficit Links Altered Network Activity and Cognitive Dysfunction in Alzheimer Model. *Cell* 149, 708–721. <https://doi.org/10.1016/j.cell.2012.02.046>.

Olah, V.J., Goettmoeller, A.M., Dimidschstein, J., and Rowan, M.J. (2021). Biophysical Kv channel alterations dampen excitability of cortical PV interneurons and contribute to network hyperexcitability in early Alzheimer's.

<https://www.biorxiv.org/content/10.1101/2021.10.25.465789v1>

-Cell-type-specific physiological changes in models of amyloid pathology certainly evolves as a function of age (disease-stage). For example, neuronal excitability changes are apparent in cortical circuits from pre-plaque hAPP mice. Thus the authors should remove characterizations of their model as 'early', as it likely better models intermediate-late stage of AD. The authors should also better discuss how their findings (here in 'older' post-plaque) stage fit with other studies using younger mice.

-What are the expected baseline firing rates of PV, Sst, and PCs from the 'ground truth' in vivo electrophysiological recordings expected for anesthetized mice? How do these values square with those reported here using the deconvoluted Ca²⁺ events? In following, are the number of inactive cells in wild type mice higher than expected with this approach, i.e., ~60% of Sst cells are seen as inactive here. Integration of the reported values with those previously reported (for wild type mice) in the literature would strengthen the authors claims.

-Reports show that Cre expression in PV-Cre mice is potentially incomplete (Nigro et al. 2021). The authors should acknowledge how might this effect their data.

-In the discussion, the authors state that 'Hyperactive SOM interneurons are the most likely driver of this dyshomeostasis since they supply aberrant inhibition to layer 2/3 pyramidal cells, thus preventing firing in excitatory cells^{25,47,48}'. This statement is not necessarily supported by the data. In fact, changes in PV cell activity could in theory have a greater effect on AP firing in excitatory cells within the local circuit, in which case one would predict hyperexcitability in postsynaptic pyramidal neurons. An alternative explanation is that intrinsic changes to excitatory cells result in their reduced activity rates. The authors could improve the manuscript by expanding their interpretations to include these, and other potential cellular and circuit mechanisms in context.

Minor points:

-Introduction, 'Thus, it remains unclear if network alterations in AD are due to increased or decreased excitability of excitatory cells'. Potentially false dichotomy, as changes to PC excitability may not necessarily be required (for example inhibitory cells alone could cause global changes).

-Introduction 'For instance, hippocampal PV interneurons are hypoactive in amyloidosis mouse models^{28,29}'. Not always- need to be more careful, PV cell hyperexcitability seen in hippocampus in some reports.

-In the results, the authors should remind the reader which cortical region Ca²⁺ imaging was being performed in.

We thank the reviewers for their careful reading of our manuscript, valuable comments and suggestions to improve our manuscript. We have a detailed point-by point response to the questions as follows and highlighted our responses:

Reviewer #1 (Remarks to the Author):

In this manuscript, the authors used cross-breeding of the APP/PS1 mouse model of Alzheimer's Disease with interneuron-specific lines to examine spontaneous activity changes of excitatory and PV and SST inhibitory neuron classes under isoflurane anesthesia. They conclude that SST interneurons are 'hyperactive', whereas pyramidal neurons and PV interneurons are 'hypoactive'.

Results are reported exclusively from deconvolved calcium transients. This is not convincing. They first need to perform simultaneous patch recordings from the GCaMP expressing excitatory and inhibitory neurons for a ground truth calibration of their calcium signals that then can inform the deconvolution to estimate the underlying spike rates. The lack of ground truth data is a significant problem of the manuscript because all conclusions rely on this approach (which is also not well described and not validated). Another major limitation is that even if the results hold, there is a lack of mechanistic insight; mainly, how the cell type specific changes are causally linked must be worked out before publication.

1- Patch clamp experiments require extensive technical experience that is not currently available in our department. Our lab would need several months of training to acquire these skills and utilize them to calibrate and optimize GCaMP data analysis. Additionally, simultaneous patch-clamp recordings have been routinely performed in GCaMP articles and successfully validated the data generated using GCaMP indicators (see Chen et al., 2013; Dana et al., 2019, see also Pachitariu et al., 2018). A study in the Journal of Neuroscience by Pachitariu et al. 2018 compared several analysis algorithms in a dataset of mutual calcium imaging and ground-truth electrophysiology recordings. The article compared supervised methods involving training the calcium data with ground truth data and non-supervised methods. The article concluded that the non-negative deconvolution algorithm (implemented in our study) outperformed all other algorithms, including supervised methods. Therefore, using a supervised method will not necessarily improve the accuracy of our analysis. Our results will likely underestimate actual firing rates because of the relatively slow calcium indicator responses to spiking and our conservative thresholded-deconvolution approach. These details are now mentioned at the start of the results section.

To compare our findings to previous GCaMP studies in AD models and to show that the analysis parameters do not significantly impact the manuscript findings, we have recalculated calcium event rates for each cell type (Reviewer Figure1) using a simple noise thresholding approach (see Busche et al. 2019 Nature Neuroscience and Korzhova et al. 2021 Communications Biology). This method identifies peaks with intensities three standard deviations above the noise level without accounting for the well-understood calcium indicator response kinetics to action potentials. Despite the limitations of these simple methods, we still show similar results for almost all cell types except for PV cells which did not reach statistical significance when the analysis was done on mice (Reviewer Figure1).

None of those mentioned above studies have calibrated their GCaMP recordings with in vivo or in vitro electrophysiology measurements.

We understand that a calibration using simultaneous patch-clamp recording and multiphoton imaging of GCaMP-expressing cells would allow us to optimize the deconvolution parameters and estimate cell-specific firing rates more precisely. However, this limitation should not affect the conclusion of this manuscript since the same analysis method was consistently applied across all groups. Additionally, genetically encoded calcium indicators are the only non-invasive tool to measure cell-type firing activity from several cells and quantify the number of inactive cells in vivo. Therefore, despite these limitations, the study findings are a novel and essential addition to the existing literature. Because we understand that the inferred event rates are not identical to electrophysiological-measured firing rates, we refrained from using the term "firing rates" throughout the manuscript and used "event rates" instead.

Understanding the causal link between interneurons subtypes and excitatory neurons is not trivial and would require several experimental approaches beyond this study's scope. We refrained from suggesting any mechanistic links in our title and throughout the discussion since we understand the limitations of our study. Future work will include targeting interneurons using chemogenetic and optogenetic approaches, observing these manipulations' effect on excitatory neurons at extended timepoints, and using different light stimulation durations or chemogenetic ligand concentrations.

- 2- The anesthesia limitation has been addressed at the last segment of our discussion section (highlighted in the revised manuscript). We chose to perform our experiments under anesthesia for compatibility with previous reports of neuronal hyperactivity. Additionally, while awake imaging is physiologically relevant, it is often complicated by variability due to brain state and may not necessarily lead to a clear conclusion. The data acquisition in our current study lasted for around one year between animal breeding and aging, surgery, and imaging. It is not feasible to repeat the whole study at different brain states for this manuscript submission.

Figure 1 b/c, top panel - please clarify how these images were generated (for all the subsequent in vivo images). There is almost no background, and GCaMP-expressing neurons are saturated and have a filled nucleus which can be a sign of toxicity.

- 3- We use fast imaging at low resolution to better sample the GCaMP response to neuronal activity. To improve signal-to-noise ratio in our representative images, we averaged 500- images from a time-lapse recording of each cell type at 256x256 resolution. We have added these details to the figure captions of Figures 1-3. Our interneuron recordings have a significantly lower backgrounds relative to excitatory cells since interneuron fields of view contain fewer cells. Because the background is much dimmer than somas in interneuron recordings, it is rendered invisible after adjusting the image contrast to visualize all cells. We also noticed that, unlike excitatory cell recordings, the unique cytoplasmic localization of GCaMP is not always obvious in interneuron recordings. Other imaging studies using GCaMP have reported similar images. Please see figures 2 and 3 in Adler et al. *Neuron* 102, 202–216, April 3, 2019. Also, see Figures 1B and 3A in Arriaga M et al. *J Neurosci.* 2017. 20;37(38):9222-9238. We have attached mean images of Movie2 (PV interneurons), which show soma -filled cells similar to our representative images despite all cells being active (Reviewer Figure 2). Updated representative images are included in

the revised manuscript, and we will be ready to modify them to any recommended appropriate format, if needed.

Figure 1 b/c, bottom panel - please clarify why the example neurons have different noise levels although they are from the same recording? This is also an issue in several of the subsequent example traces.

- 4- Representative traces were not from the same recording in the older manuscript version. We realize that this could be confusing for the reader, thus we have added new representative images with matched traces in all the figures of the revised version of that manuscript.

Figure 4 - Please clarify how this analysis was done and show data for the biological replicates (i.e., animals). The strong significance is because neurons from all animals were pooled in the analysis. Does it appear that much of the effect is driven by an increase in pairwise correlations at 210 μ m in WT (see panel b)? This is puzzling to me as one would not expect an increase in correlation with distance?

- 5- The analysis for Figure 4 was done from all pairwise correlations in the old version of the manuscript. We wanted to understand the effect of event rates and inter-neuronal distances on pairwise correlation values. Pairwise neuronal correlation values are highly variable and a single correlation value per animal would make the interpretation of these relationships difficult. We, therefore, used all pairwise correlation values to smooth this variability and reduce the error bars in the older version of the manuscript. Chen et al. used a similar model to study neuronal correlations (Nature Neuroscience 23, 520–532 (2020)). Nevertheless, for clarity, we have reanalyzed data in Figure 4 so that the data are presented with mean values from each animal and moved the analysis per pairwise correlations to Supplementary Figure5. The two analyses lead to very similar conclusions (i.e. a strong trend towards decreased pairwise correlation in excitatory cells, a significant main interneuronal- distance effect, and a significant main event rate effect on excitatory pairwise correlations). Only 6% of the pairwise correlations were at 210 μ m in both genotypes, so there is a higher chance of error. To make sure that our results are not driven by this particular datapoint, we recalculated the average pairwise correlation values for WT and APP CAMK neurons after excluding pairwise correlations at 210-time bin (Reviewer Figure3). It doesn't seem that the effects are exclusively driven by this particular data point (Reviewer figure 3).

"We propose a mechanism by which amyloid plaque-driven hyperactivity in SOM interneurons leads to decreased firing of pyramidal cells." This is a misleading statement because no mechanistic link has been established.

- 6- We replaced this statement with the following in the revised manuscript "We provide evidence for the failure of homeostatic mechanisms to maintain appropriate neuronal firing in APP/PS1 mice. Because the hyperactivity of SOM interneurons correlates with proximity to amyloid plaques SOM interneurons can potentially drive this dyshomeostasis since they supply aberrant inhibition to layer 2/3 pyramidal cells^{27,58,59}. SOM interneurons were also reported to strongly inhibit PV interneurons in cortical layers 2/3 during visual processing⁶⁰. SOM interneuron hyperactivity could thus suppress PV interneuron firing in APP/PS1 mice. Indeed, we saw hypoactivity within PV interneurons."

"It is important to note that the hyperactivity phenotype has been observed in amyloidosis mouse models exhibiting a considerable number of seizures, and it remains unknown whether this phenotype occurs in amyloidosis models lacking the seizure phenotype." The statement suggests that this study addresses this

issue; however, they use APP/PS1 mice known to have seizures. Indeed, they refer to a JAX datasheet that the APP/PS1 line on this particular background does not have seizures; however, I am not convinced they can rule out the presence of seizures in their cohort. They need to demonstrate the absence of seizures electrographically in their cohort if they wish to make such claims (it is, for example, known that animal housing affects the incidence of seizures).

7- Since we did not have the data to support the complete absence of seizures in APP/PS1 mice on Ch3 background, we removed the following paragraph including the abovementioned statement from the introduction.

“It is important to note that the hyperactivity phenotype has been observed in amyloidosis mouse models exhibiting a considerable number of seizures^{20,21}, and it remains unknown whether this phenotype occurs in amyloidosis models lacking the seizure phenotype. Answering this question is essential to understanding whether the hyperactivity phenotype is related to amyloid plaque accumulation or plaque unrelated factors. Thus, it remains unclear if network alterations in AD are due to increased or decreased excitability of excitatory cells”

"It is possible that most of the previously reported hyperactive cells were, in fact, SOM interneurons." This is unlikely since the number of SST neurons is much smaller than the number of hyperactive neurons reported in the previous literature, as far as I can ascertain.

8- We have removed this statement from the discussion section

Reviewer #2 (Remarks to the Author):

Summary:

In the manuscript ‘Hyperactive somatostatin interneurons near amyloid plaque and cell- type-specific firing deficits in a mouse model of Alzheimer’s disease’, Algamal et al. explore changes in the activity of distinct inhibitory and excitatory neuron types in somatosensory cortex layer 2/3, in the well characterized APP/PS1 AD mouse model. Cell-type-specific differences and vulnerabilities in AD are becoming more apparent and appreciated as part of the disease process, thus this manuscript explores an important and timely topic which will be of wide interest to the field. Data from excitatory cells and two distinct interneuron types, as well as their spatial relationships to amyloid plaque deposits, makes for interesting comparisons. Generally, the manuscript is well organized, the methods are sound, and data is compelling. However several issues, mostly regarding interpretations and contextualization, should be addressed:

Major points:

-In multiple instances in the abstract and introduction, the authors state that early studies did not discriminate between neuronal subtypes (e.g.; ‘the impact of amyloid pathology on distinct neuronal types remains unexplored within intact neuronal circuits’; ‘several studies have assessed the firing activity of distinct neuronal cell types in amyloidosis mouse models’.) This is not fully accurate- several groups have explored physiological changes in several different AD models explicitly in distinct neuronal types (e.g., ex vivo in Petrache et al., 2019, Hijazi et al., 2019, Park et al., 2020, Verret et al., 2012, Olah et al. 2022 and in vivo, e.g., Bai et al. 2017, Martinez-Losa et al. 2018, Hanson et al. 2020). Throughout the results and manuscript in general, the authors should better frame the novelty of their work in the context of these studies (full refs here):

Bai, Y., Li, M., Zhou, Y., Ma, L., Qiao, Q., Hu, W., Li, W., Wills, Z.P., and Gan, W.-B. (2017). Abnormal dendritic calcium activity and synaptic depotentiation occur early in a mouse model of

Alzheimer's disease. *Mol Neurodegeneration* 12, 1–15. <https://doi.org/10.1186/s13024-017-0228-2>.

Hanson, J.E., Ma, K., Elstrott, J., Weber, M., SAILLET, S., Khan, A.S., Simms, J., Liu, B., Kim, T.A., Yu, G.-Q., et al. (2020). GluN2A NMDA Receptor Enhancement Improves Brain Oscillations, Synchrony, and Cognitive Functions in Dravet Syndrome and Alzheimer's Disease Models. *Cell Reports* 30, 381–396.e4. <https://doi.org/10.1016/j.celrep.2019.12.030>.

Hijazi, S., Heistek, T.S., Scheltens, P., Neumann, U., Shimshek, D.R., Mansvelder, H.D., Smit, A.B., and Kesteren, R.E. van (2019). Early restoration of parvalbumin interneuron activity prevents memory loss and network hyperexcitability in a mouse model of Alzheimer's disease. *Mol Psychiatry* 1–19. <https://doi.org/10.1038/s41380-019-0483-4>.

Martinez-Losa, M., Tracy, T.E., Ma, K., Verret, L., Clemente-Perez, A., Khan, A.S., Cobos, I., Ho, K., Gan, L., Mucke, L., et al. (2018). Nav1.1-Overexpressing Interneuron Transplants Restore Brain Rhythms and Cognition in a Mouse Model of Alzheimer's Disease. *Neuron* 0. <https://doi.org/10.1016/j.neuron.2018.02.029>.

Park, K., Lee, J., Jang, H.J., Richards, B.A., Kohl, M.M., and Kwag, J. (2020). Optogenetic activation of parvalbumin and somatostatin interneurons selectively restores theta-nested gamma oscillations and oscillation-induced spike timing-dependent long-term potentiation impaired by amyloid β oligomers. *BMC Biol* 18, 1–20. <https://doi.org/10.1186/s12915-019-0732-7>.

Petrache, A.L., Rajulawalla, A., Shi, A., Wetzell, A., Saito, T., Saido, T.C., Harvey, K., and Ali, A.B. (2019). Aberrant Excitatory–Inhibitory Synaptic Mechanisms in Entorhinal Cortex Microcircuits During the Pathogenesis of Alzheimer's Disease. *Cereb Cortex* 29, 1834–1850. <https://doi.org/10.1093/cercor/bhz016>.

Verret, L., Mann, E.O., Hang, G.B., Barth, A.M.I., Cobos, I., Ho, K., Devidze, N., Masliah, E., Kreitzer, A.C., Mody, I., et al. (2012). Inhibitory Interneuron Deficit Links Altered Network Activity and Cognitive Dysfunction in Alzheimer Model. *Cell* 149, 708–721. <https://doi.org/10.1016/j.cell.2012.02.046>.

Olah, V.J., Goettmoeller, A.M., Dimidschstein, J., and Rowan, M.J. (2021). Biophysical Kv channel alterations dampen excitability of cortical PV interneurons and contribute to network hyperexcitability in early Alzheimer's. <https://www.biorxiv.org/content/10.1101/2021.10.25.465789v1>

- 1- We thank the reviewer for providing us with references to these important studies. We have reframed our manuscript to provide a better context for our study novelty. To the best of our knowledge, studies that discriminate between firing activity of different cell-types were done on ex-vivo brain slices or using non-discriminatory in vivo population recordings such as EEG (including Martinez-Losa et al. 2018, Hanson et al. 2020). On the other hand, in vivo single-cell imaging studies such as Bai et al. Busch et al. did not discriminate between cell-type-specific firing in vivo. Therefore, to the best of our knowledge, this is the first study to monitor single-cell firing of distinct cell-types in vivo during spontaneous activity.
 - a. in the abstract, the following sentence ‘the impact of amyloid pathology on distinct neuronal types remains unexplored within intact neuronal circuits’ was replaced with ‘‘However, the impact of amyloid pathology on the spontaneous activity of distinct neuronal types remains unexplored **in vivo**’’.
 - b. Introduction: ‘‘several studies have assessed the firing activity of distinct neuronal cell types in amyloidosis mouse models’’ was replaced with ‘‘Several studies have assessed

single-cell neuronal activity in amyloidosis mouse models **in ex-vivo brain slices 16–19 and in vivo**^{5,20,21}. **While ex-vivo studies have monitored cell-type-specific neuronal activity, in vivo** studies did not discriminate between neuronal subtypes and mainly reported the activity of excitatory neurons as they represent 80-90% of cortical neurons²². These reports have provided mixed findings of aberrant neuronal hyperactivity^{5,6} and hypoactivity^{5,21,23, 24,25}

- c. Results: “Studies examining indiscriminate neuronal populations have shown that neurons near amyloid plaques at distances less than 60 μm are more active than those farther away⁵.” Was changed to “Studies examining spontaneous activity in indiscriminate neuronal populations in vivo have shown that neurons near amyloid plaques at distances less than 60 μm are more active than those farther away⁵.”
- d. Results: “We then asked whether this hyperactivity can be generalized to other interneuron subtypes in APP/PS1 mice” replaced with “We then asked whether **the spontaneous hyperactivity in the somatosensory cortex** can be generalized to other interneuron subtypes in APP/PS1 mice **in vivo**.”
- e. Results: Because of the numerous reports of hyperactive cells in APP/PS1 mice, we hypothesized that excitatory cells are hyperactive as result of a dominant hypoactivity in PV interneurons was replaced with “Because of the numerous reports of hyperactive cells in APP/PS1 mice **in vivo**, we hypothesized that excitatory cells are hyperactive as result of a dominant hypoactivity in PV interneurons
- f. Results: To evaluate the impact of amyloid accumulation on firing synchrony of different cell types, we calculated the Pearson correlation values (ρ) for all possible pairwise combinations of active neurons” was replaced with “To evaluate the impact of amyloid accumulation on firing synchrony of different cell types, we calculated the Pearson correlation values (ρ) for all possible pairwise combinations of active neurons **in the somatosensory cortex**.”
- g. Discussion: “Recent reports suggest that inhibitory circuits are particularly vulnerable, resulting in E/I imbalance and circuit-level dysfunction in AD^{19,28,34,43}. However, the exact nature of the disrupted inhibitory activity is not yet fully understood” was replaced with Recent reports suggest that inhibitory circuits are particularly vulnerable, resulting in E/I imbalance and circuit-level dysfunction in AD^{19,28,34,43}. However, the exact nature of the disrupted inhibitory activity is not yet fully understood **in vivo**.
- h. Discussion: “We report cell-type-specific interneuron dysfunction in APP/PS1 mice.” was replaced with “We report cell-type-specific interneuron dysfunction **during spontaneous activity recorded in vivo** in APP/PS1 mice.

-Cell-type-specific physiological changes in models of amyloid pathology certainly evolves as a function of age (disease-stage). For example, neuronal excitability changes are apparent in cortical circuits from pre-plaque hAPP mice. Thus the authors should remove characterizations of their model as ‘early’, as it likely better models intermediate-late stage of AD. The authors should also better discuss how their findings (here in ‘older’ post-plaque) stage fit with other studies using younger mice.

2- We now refer to our model as an early to intermediate stage AD in the revised manuscript.

What are the expected baseline firing rates of PV, Sst, and PCs from the ‘ground truth’ in vivo electrophysiological recordings expected for anesthetized mice? How do these values square with those reported here using the deconvoluted Ca²⁺ events? In following, are the number of inactive cells in wild type mice higher than expected with this approach, i.e., ~60% of Sst cells are seen as inactive here. Integration of the reported values with those previously reported (for wild type mice) in the literature would strengthen the authors claims.

3- The methods used to calculate firing rates in electrophysiological recordings varies between articles, but a rough estimate of cell-type-specific firing rates under anesthesia would be 0^b-2^{ad} Hz for SOM, 40 Hz for PV^a, and 2^a-4^c Hz for PC (see references a-d at the end). We report 0.032(0.065) Hz for SOM, 0.18 (0.19) Hz for PV and 0.068 (0.09) Hz for PC. The values between brackets represent the rates after excluding inactive cells. Our values show a clear underestimation compared to the measured electrophysiologic values but also similar relative rates, with PV rates being the fastest, followed by PC and SOM. Two factors could account for this underestimation, the nature of calcium indicator response to spiking and the methodological differences in calculating firing rates between our study electrophysiology recordings. First, calcium indicators' responses are slow (seconds) relative to spiking events (milliseconds), and while we use a deconvolution method to achieve the best estimate, we are likely to miss some events particularly during burst firing. We also use a relatively high noise threshold (2.5) and long-time decay constant (1.5 seconds) to ensure that noisy traces do not drive our conclusions. Second, most electrophysiology estimated firing rates often involve a stimulus to initiate spiking followed by spike counting on a short-time window, while in our study, we only count spontaneously occurring events over minutes with no stimulation. We could not find any reports of the number of SOM cells under anesthesia using calcium imaging or in vivo electrophysiology measurements. A Nature article by Adesnik et al. in 2012 used in vivo extracellular unit recording to calculate the maximal visually evoked firing rates of different cell types under anesthesia and during running. The article report that SOM firing rates were reduced by 15 -20 folds under anesthesia relative to running, while PV and PC remained unaffected by anesthesia. Therefore, the increased fraction of inactive cells in SOM recording could be related to their increased sensitivity to anesthesia.

- a. Adesnik, H., Bruns, W., Taniguchi, H., Huang, Z. J., & Scanziani, M. (2012). A Neural Circuit for Spatial Summation in Visual Cortex. *Nature*, *490*(7419), 226. <https://doi.org/10.1038/NATURE11526>
- b. Gentet, L. J., Kremer, Y., Taniguchi, H., Huang, Z. J., Staiger, J. F., & Petersen, C. C. H. (2012). Unique functional properties of somatostatin-expressing GABAergic neurons in mouse barrel cortex. *Nature Neuroscience*, *15*(4), 607–612. <https://doi.org/10.1038/nn.3051>
- c. Klee, J. L., Kiliaan, A. J., Lipponen, A., & Battaglia, F. P. (2020). Reduced firing rates of pyramidal cells in the frontal cortex of APP/PS1 can be restored by acute treatment with levetiracetam. *Neurobiology of Aging*, *96*, 79–86. <https://doi.org/10.1016/j.neurobiolaging.2020.08.013>
- d. Urban-Ciecko, J., & Barth, A. L. (2016). Somatostatin-expressing neurons in cortical networks. *Nature Reviews Neuroscience*, *17*(7), 401–409. <https://doi.org/10.1038/nrn.2016.53>

-Reports show that Cre expression in PV-Cre mice is potentially incomplete (Nigro et al. 2021). The authors should acknowledge how might this effect their data.

4- The expression efficiency appears to be limited in the perirhinal cortex and not in the barrel cortex of PV-Cre mice, as described by Nigro et al.2021. The decrease in expression efficiency can potentially affect studies counting the number of PV cells at different experimental conditions. However, this should not affect functional studies aiming to understand the cellular properties of PV cells, like our study, because Nigro et al. also showed that the specificity of expression to PV is close to 90% in all regions (Nigro et al., 2021). Our supplementary figure1 further corroborates this showing approximately 86% specificity for PV cells in PV-Cre mice.

-In the discussion, the authors state that ‘Hyperactive SOM interneurons are the most likely driver of this dyshomeostasis since they supply aberrant inhibition to layer 2/3 pyramidal cells, thus preventing firing in excitatory cells 25,47,48’. This statement is not necessarily supported by the data. In fact, changes in PV cell activity could in theory have a greater effect on AP firing in excitatory cells within the local circuit, in which case one would predict hyperexcitability in postsynaptic pyramidal neurons. An alternative explanation is that intrinsic changes to excitatory cells result in their reduced activity rates. The authors could improve the manuscript by expanding their interpretations to include these, and other potential cellular and circuit mechanisms in context.

- 5- We have added the following statements to our discussion section to discuss other interpretations of our findings.
 - a. Hence, neuronal circuits will respond to changes in firing activity, particularly in excitatory cells, with compensatory mechanisms to restore the perturbed firing homeostasis^{56,57}. The process of restoring firing homeostasis involves two well-documented mechanisms. One, the circuit can adjust synaptic strength by gradually changing the ratio of inhibitory and excitatory synapses in the direction that normalizes the perturbation, or two, it can modify the intrinsic excitability to balance synaptic input and firing rates⁵⁶.
 - b. We provide evidence for the failure of homeostatic mechanisms to maintain appropriate neuronal firing in APP/PS1 mice. Because the hyperactivity of SOM interneurons correlates with proximity to amyloid plaques SOM interneurons can potentially drive this dyshomeostasis since they supply aberrant inhibition to layer 2/3 pyramidal cells^{27,58,59}.
 - c. “In addition to the synaptic mechanisms, alterations in the intrinsic excitability of different cell types can account for the changes in their activity patterns⁵⁶. While the evidence for altered intrinsic excitability of excitatory cells in amyloidosis mouse models has been inconsistent^{19,61,62}, a decrease in intrinsic excitability of pyramidal cells could potentially account for their hypoactivity in APP/PS1 mice.”.

Minor points:

-Introduction, ‘Thus, it remains unclear if network alterations in AD are due to increased or decreased excitability of excitatory cells’. Potentially false dichotomy, as changes to PC excitability may not necessarily be required (for example inhibitory cells alone could cause global changes).

- 6- Since we did not have the data to support the complete absence of seizures in APP/PS1 mice on Ch3 background, we removed the following paragraph including the abovementioned statement from introduction in the new version of the manuscript.

“It is important to note that the hyperactivity phenotype has been observed in amyloidosis mouse models exhibiting a considerable number of seizures^{20,21}, and it remains unknown whether this phenotype occurs in amyloidosis models lacking the seizure phenotype. Answering this question is essential to understanding whether the hyperactivity phenotype is related to amyloid plaque accumulation or plaque unrelated factors. Thus, it remains unclear if network alterations in AD are due to increased or decreased excitability of excitatory cells”

-Introduction ‘For instance, hippocampal PV interneurons are hypoactive in amyloidosis mouse models^{28,29}’. Not always- need to be more careful, PV cell hyperexcitability seen in hippocampus in

some reports.

7- We have modified the sentence to include the differences in PV activity that have been reported in the literature. The following sentence “For instance, hippocampal PV interneurons are hypoactive in amyloidosis mouse models^{28,29} was replaced with “For instance, hypoactivity^{17,18,32–35} and hyperactivity¹⁹ of PV interneurons were reported in several AD mouse models of amyloidosis.

-In the results, the authors should remind the reader which cortical region Ca²⁺ imaging was being performed in.

8- We have added the imaging region to all paragraphs in the results section.

Reviewers' comments:

Reviewer #1 (Remarks to the Author):

I'd like to thank the authors for their responses. My concerns have been addressed with additional analyses and text/figure modifications. I note that some of the previously reported main results (e.g., parvalbumin firing rates and pairwise correlations) are no longer statistically significant after re-analysis. The remaining main results are that somatostatin interneurons show increased while excitatory neurons show reduced spontaneous activity in anesthetized APP/PS1 mice. The mechanisms and relevance of these data remain unexplored.

Specific concerns:

'Reviewer figures' - I do not understand why the 'reviewer figures' are have not been included in the manuscript? They provide important additional information.

Figure 2, and text: 'PV interneurons are hypoactive in APP/PS1 mice'. This is not convincing since the new analysis of calcium transients (which is their primary readout) does not show any significant differences between groups.

Figure 1b,c - In the new representative traces it seems that the noise in the WT recordings is much (ie, 2-3 fold) higher compared to the APP/PS1 mice. This is a concern and potential confound as a lower signal-to-noise ratio in WT mice could lead to an underestimation of spontaneous firing. This needs clarification.

Figure 3c - I still don't understand why the noise level between traces is so different if they come from the same recordings (see, e.g., cell 1 vs cell 3)? This is puzzling.

Figure 4 - This data are now not significant anymore and should be reported accordingly (not as: 'there is a strong trend for a...'). More generally, I do not see the point to show this as a main figure when there is no statistically significant difference and it is unclear how it adds to current knowledge.

The discussion section is longer than the results section and too speculative in view of the limited data that are reported here.

Reviewer #2 (Remarks to the Author):

Summary:

In the manuscript 'Hyperactive somatostatin interneurons near amyloid plaque and cell-type-specific firing deficits in a mouse model of Alzheimer's disease', Algamal et al. explore changes in the activity of distinct inhibitory and excitatory neuron types in somatosensory cortex layer 2/3, in the well characterized APP/PS1 AD mouse model. Cell-type-specific differences and vulnerabilities in AD are becoming more apparent and appreciated as part of the disease process, thus this manuscript explores an important and timely topic which will be of wide interest to the field. Data from excitatory cells and two distinct interneuron types, as well as their spatial relationships to amyloid plaque deposits, makes for interesting comparisons. Both reviewers generally agreed that some technical and interpretation limits of the original paper needed to be addressed/discussed.

Evaluation:

- Overall the authors seemed to adequately address all points raised in the initial review.
- The authors improved their analysis methodology for Ca²⁺ imaging with a more standardized/unbiased approach, however could not perform the 'ground truth' electrophysiological analysis as requested in the initial review. Despite this, the improved analysis in addition to the qualifying language in the results regarding relative changes in activity seem to

be suitable. Authors could discuss that future complementary studies using MEAs in unanaesthetized animals will be quite important and necessary to fully confirm these neuron-type-specific findings in vivo.

- Despite the more rigorous analysis, the overall quite interesting cell-type-specific results remained the same. Importantly, the PV and PC findings were still in line with expectations from the literature as a whole from 'APP' mice, although was the first demonstration to my knowledge of simultaneous in vivo confirmation. This is an important finding as future experiments can now test the hypothesis that PV hypoexcitability is causally related to overall circuit hyperexcitability. This in addition to the very novel finding on SOM interneurons.

- I would not expect the authors to perform a 'mechanistic' evaluation for the effects seen. This is outside of the scope of the paper in my opinion and would need to double or triple the size of the manuscript with several ex vivo recordings. On the contrary, the results here provide some needed inspiration for cellular and molecular-minded neurophysiologists to go after mechanisms which would then be 'fed back' in vivo for testing.

- Because of the quite realistic concerns based on the Ca²⁺ imaging methodology and the cell-type-specific concerns regarding anesthesia in general, I would recommend the authors remove the 'hyperactive SOM interneurons' claim from the title. A more generalized title may be more appropriate, i.e., "distinct neuron-type-specific activity deficits..."

- Furthermore because of the potential importance of cell-type-specific effect of anesthesia (Adesnik et al. in 2012) authors should include 'under anesthesia' somewhere in the abstract.

We thank the reviewers for their valuable comments and suggestions. We believe their efforts have helped us improve our manuscript significantly. We have a detailed point-by-point response to the questions as follows and highlighted our responses.

Reviewer #1 (Remarks to the Author):

I'd like to thank the authors for their responses. My concerns have been addressed with additional analyses and text/figure modifications. I note that some of the previously reported main results (e.g., parvalbumin firing rates and pairwise correlations) are no longer statistically significant after re-analysis. The remaining main results are that somatostatin interneurons show increased while excitatory neurons show reduced spontaneous activity in anesthetized APP/PS1 mice. The mechanisms and relevance of these data remain unexplored.

Specific concerns:

'Reviewer figures' - I do not understand why the 'reviewer figures' are have not been included in the manuscript? They provide important additional information.

We have added the reviewer figures as supplementary figures.

Figure 2, and text: 'PV interneurons are hypoactive in APP/PS1 mice'. This is not convincing since the new analysis of calcium transients (which is their primary readout) does not show any significant differences between groups.

We believe that the original analysis (presented in the main figures) using thresholded deconvolution is more accurate (see Pachitariu et al. J Neuroscience, 2018 and Chen et al. Nature Neuroscience, 2020). Therefore, we used it to deduce our conclusions. It would be misleading to state that PV activity is similar in WT and APP mice giving the data presented in the main manuscript figures using the more informed deconvolution method and the cumulative distribution of event rates in PV cells using the less accurate calcium event counting method (Supplementary Figure 6d).

Figure 1b,c - In the new representative traces it seems that the noise in the WT recordings is much (ie, 2-3 fold) higher compared to the APP/PS1 mice. This is a concern and potential confound as a lower signal-to-noise ratio in WT mice could lead to an underestimation of spontaneous firing. This needs clarification.

The noise level does not seem to be 2-3-fold higher in all cells of WT-SOM relative to APP/PS1-SOM. For instance, noise appears similar in inactive cells (see superimposed traces in Reviewer Figure1_b, particularly cells 3 and 4), but we understand that cell #1 in WT-SOM may confuse. Unlike electrophysiology measurement, where voltage is a final readout, in calcium imaging studies, the final readout is delta F/F, i.e., the change in fluorescence relative to baseline fluorescence. The concentration of GCaMP that gets ultimately expressed in different cells and animals is not identical. Therefore, cells will express different GCaMP concentrations within the same recording and appear with variable brightness and variable baseline noise. After calculating the delta f/f values, dimmer inactive cells may have higher apparent noise. This could be the case for cell#1 in WT animals which appear slightly dimmer and noisier. We discussed this limitation in lines 250-254 of the manuscript. Variable noise levels in the same

GCaMP recording are not uncommon; please see figures 3d and 4b in Busche et al. Nature Neuroscience 2019 and Figure 1c in Korzhova et al. Communications Biology, 2021. Figure 3c - I still don't understand why the noise level between traces is so different if they come from the same recordings (see, e.g., cell 1 vs cell 3)? This is puzzling.

Please see our answer to the previous question. It is hard to control the concentration of GCaMP that gets ultimately expressed in different cells. Additionally, the brightness and baseline noise change could be related to intrinsic cellular differences in basal calcium levels or calcium buffering capacity. After calculating the $\Delta f/f$ values, dimmer inactive cells may appear noisier.

Figure 4 - This data are now not significant anymore and should be reported accordingly (not as: 'there is a strong trend for a...'). More generally, I do not see the point to show this as a main figure when there is no statistically significant difference and it is unclear how it adds to current knowledge.

It is very important to understand whether the changes in event rates are accompanied by network dyssynchronization in APP/PS1 mice. The lack of dyssynchronization despite altered event rates is an important finding and adds to our current understanding of neuronal network function in Alzheimer's disease. Although the golden p-value of 0.05 was not met, it is also important to show that this data was almost significant in case future replication studies report otherwise. Regardless of the significance, the positive association between pairwise correlation values and inter-neuronal distances and event rates is an important validation of the ability of calcium imaging to replicate well-established electrophysiological phenomena and should be of interest to the readers. It also provides a platform of analysis that can be followed in future calcium and electrophysiology studies. Negative data is not necessarily trivial.

The discussion section is longer than the results section and too speculative in view of the limited data that are reported here.

Please note that upon the request of Dr. C. Justin Lee, we have added a discussion point on tonic inhibition as he believes that it can potentially explain our findings in this particular mouse model. To shorten our discussion, we have removed the following paragraph:

Several *in vitro* studies have shown that the application of amyloid-beta oligomers induces neuronal hyperexcitability through several mechanisms that involve blocking glutamate uptake leading to glutamate spillover²³, reduction of endocannabinoid-mediated disinhibition³², and increasing neuronal resting membrane potential⁵⁰. A question that emerges from the current study is how amyloid beta-induced hyperactivity can be observed in SOM interneurons only and not in PV or excitatory cells. To answer this intriguing question, we adopt a network model in which two sources can perturb each neuronal cell-type activity: nearby amyloid plaque and excitatory and inhibitory network inputs. For instance, despite being conflicted by nearby amyloid plaque, excitatory neurons are hypoactive in APP/PS1 mice due to the hyperactive SOM inhibitory inputs. These opposite perturbations of excitatory cells can also explain the lack of correlation between excitatory cell activity and the distance to amyloid plaque.

Reviewer #2 (Remarks to the Author):

Summary:

In the manuscript 'Hyperactive somatostatin interneurons near amyloid plaque and cell-type-specific firing deficits in a mouse model of Alzheimer's disease', Algamal et al. explore changes in the activity of distinct inhibitory and excitatory neuron types in somatosensory cortex layer 2/3, in the well characterized APP/PS1 AD mouse model. Cell-type-specific differences and vulnerabilities in AD are becoming more apparent and appreciated as part of the disease process, thus this manuscript explores an important and timely topic which will be of wide interest to the field. Data from excitatory cells and two distinct interneuron types, as well as their spatial relationships to amyloid plaque deposits, makes for interesting comparisons. Both reviewers generally agreed that some technical and interpretation limits of the original paper needed to be addressed/discussed.

Evaluation:

- Overall the authors seemed to adequately address all points raised in the initial review.
- The authors improved their analysis methodology for Ca²⁺ imaging with a more standardized/unbiased approach, however could not perform the 'ground truth' electrophysiological analysis as requested in the initial review. Despite this, the improved analysis in addition to the qualifying language in the results regarding relative changes in activity seem to be suitable. Authors could discuss that future complementary studies using MEAs in unanaesthetized animals will be quite important and necessary to fully confirm these neuron-type-specific findings in vivo.

We address this limitation in lines 253-257 of the discussion, but we added that Multi-Electrode arrays should be used.

- Despite the more rigorous analysis, the overall quite interesting cell-type-specific results remained the same. Importantly, the PV and PC findings were still in line with expectations from the literature as a whole from 'APP' mice, although was the first demonstration to my knowledge of simultaneous in vivo confirmation. This is an important finding as future experiments can now test the hypothesis that PV hypoexcitability is causally related to overall circuit hyperexcitability. This in addition to the very novel finding on SOM interneurons

- I would not expect the authors to perform a 'mechanistic' evaluation for the effects seen. This is outside of the scope of the paper in my opinion and would need to double or triple the size of the manuscript with several ex vivo recordings. On the contrary, the results here provide some needed inspiration for cellular and molecular-minded neurophysiologists to go after mechanisms which would then be 'fed back' in vivo for testing.

We thank the reviewer for his view of our work, and we hope these findings will inspire more detailed mechanistic investigations in the future.

- Because of the quite realistic concerns based on the Ca²⁺ imaging methodology and the cell-type-specific concerns regarding anesthesia in general, I would recommend the authors

remove the 'hyperactive SOM interneurons' claim from the title. A more generalized title may be more appropriate, i.e., "distinct neuron-type-specific activity deficits..."

We have changed the title to "Reduced excitatory neuron activity and interneuron-type-specific deficits in a mouse model of Alzheimer's disease."

Please note that upon the request of Dr. C. Justin Lee, we have added a discussion point on tonic inhibition as he believes that it can potentially explain our findings in this particular mouse model. Reviewer 1 asked us to shorten our discussion, so we have removed the following paragraph:

Several *in vitro* studies have shown that the application of amyloid-beta oligomers induces neuronal hyperexcitability through several mechanisms that involve blocking glutamate uptake leading to glutamate spillover²³, reduction of endocannabinoid-mediated disinhibition³², and increasing neuronal resting membrane potential⁵⁰. A question that emerges from the current study is how amyloid beta-induced hyperactivity can be observed in SOM interneurons only and not in PV or excitatory cells. To answer this intriguing question, we adopt a network model in which two sources can perturb each neuronal cell-type activity: nearby amyloid plaque and excitatory and inhibitory network inputs. For instance, despite being conflicted by nearby amyloid plaque, excitatory neurons are hypoactive in APP/PS1 mice due to the hyperactive SOM inhibitory inputs. These opposite perturbations of excitatory cells can also explain the lack of correlation between excitatory cell activity and the distance to amyloid plaque.

- Furthermore because of the potential importance of cell-type-specific effect of anesthesia (Adesenik et al. in 2012) authors should include 'under anesthesia' somewhere in the abstract

We have added "under anesthesia" wording to the abstract.

Reviewers' comments:

Reviewer #1 (Remarks to the Author):

I'm not at all convinced that the deconvolved calcium results - without ground truth data - are superior to the analysis of the primary calcium data. They now have two conflicting results about parvalbumin activity in the AD mouse model and chose to present the significant one instead of validating this result independently. I agree that negative data is important, but it must be reported accordingly.

In sum, I am worried that the paper's conclusions might not be reproducible and advise against publication in its current form.

Reviewer #2 (Remarks to the Author):

All major issues were sufficiently addressed.

However I would argue that new lengthy discussion point is more speculative than I'm comfortable with. I'd argue to remove it (the other reviewer also thinks the discussion is too long as well). There are several other possibilities which could explain the authors cell-type-specific findings which also could be speculated on outside of this set of ideas (for which no references are included).

Reviewers' comments:

Reviewer #1 (Remarks to the Author):

I'm not at all convinced that the deconvolved calcium results - without ground truth data - are superior to the analysis of the primary calcium data. They now have two conflicting results about parvalbumin activity in the AD mouse model and chose to present the significant one instead of validating this result independently. I agree that negative data is important, but it must be reported accordingly.

In sum, I am worried that the paper's conclusions might not be reproducible and advise against publication in its current form.

We understand the reviewers concerns regarding the uncertainty of our PV findings therefore, we have added the following sentences to our results and discussion to address this issue.

Results:

Although PV interneuron activity was not significantly different between groups using peak counting (Supplementary Figure 6c), the cumulative frequency distribution of event rates was different in PV-APP interneurons relative to PV-WT interneurons (Supplementary Figure 6d). In summary, while the analysis method has a minimal effect on most of our findings, PV interneuron results are limited and should be interpreted cautiously.

Discussion:

It is important to note that our PV findings are limited and uncertain since the peak counting method did not show a significant effect. Future studies should measure PV interneuron firing rates in AD models using in vivo electrophysiological recordings to validate our findings.

We want to draw the reviewer's attention to the fact that there was a significant main genotype effect in Figure 4b, i.e., correlation values were significantly different between WT-EX and APP-EX mice. Therefore, we believe that our wording is suitable to describe this particular data panel.

Reviewer #2 (Remarks to the Author):

All major issues were sufficiently addressed.

However I would argue that new lengthy discussion point is more speculative than I'm comfortable with. I'd argue to remove it (the other reviewer also thinks the discussion is too long as well). There are several other possibilities which could explain the authors cell-type-specific findings which also could be speculated on outside of this set of ideas (for which no references are included).

We have removed the new discussion point.